# SM$^4$Depth: Seamless Monocular Metric Depth Estimation across Multiple Cameras and Scenes by One Model

Yihao Liu[*]
l1h_l1h_l1h@163.com
Beijing University of Posts and
Telecommunications
China

Feng Xue[*]
feng.xue@unitn.it
University of Trento
Italy

Anlong Ming[†]
mal@bupt.edu.cn
Beijing University of Posts and
Telecommunications
China

Mingshuai Zhao
mingshuai_z@bupt.edu.cn
Beijing University of Posts and
Telecommunications
China

Huadong Ma
mhd@bupt.edu.cn
Beijing University of Posts and
Telecommunications
China

Nicu Sebe
niculae.sebe@unitn.it
University of Trento
Italy

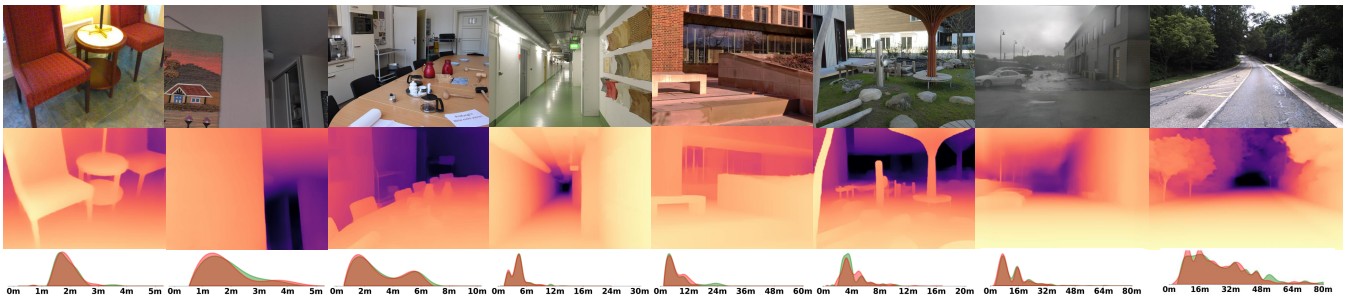

Figure 1: Depth and distribution visualization of SM$^4$Depth that enables good generalization across multiple metric depth datasets captured by different sensors. Top: input images. Middle: depth prediction. Bottom: distribution of the prediction (red) and ground truth (green). Six datasets: SUN-RGBD[48], DIODE[54], iBims-1[23], ETH3D[42], nuScenes-val[6], and DDAD[20].

## ABSTRACT

In the last year, universal monocular metric depth estimation (universal MMDE) has gained considerable attention, serving as the foundation model for various multimedia tasks, such as video and image editing. Nonetheless, current approaches face challenges in maintaining consistent accuracy across diverse scenes without scene-specific parameters and pre-training, hindering the practicality of MMDE. Furthermore, these methods rely on extensive datasets comprising millions, if not tens of millions, of data for training, leading to significant time and hardware expenses. This paper presents SM$^4$Depth, a model that seamlessly works for both indoor and outdoor scenes, without needing extensive training data and GPU clusters. Firstly, to obtain consistent depth across diverse scenes, we propose a novel metric scale modeling, i.e., variation-based unnormalized depth bins. It reduces the ambiguity of the conventional metric bins and enables better adaptation to large depth gaps of scenes during training. Secondly, we propose a "divide and conquer" solution to reduce reliance on massive training data. Instead of estimating directly from the vast solution space, the metric bins are estimated from multiple solution sub-spaces to reduce complexity. Additionally, we introduce an uncut depth dataset, BUPT Depth, to evaluate the depth accuracy and consistency across various indoor and outdoor scenes. Trained on a consumer-grade GPU using just 150K RGB-D pairs, SM$^4$Depth achieves outstanding performance on the most never-before-seen datasets, especially maintaining consistent accuracy across indoors and outdoors. The code can be found *here*.

[*]Both authors contributed equally to this research.
[†]Corresponding author.

## CCS CONCEPTS

• **Computing methodologies → 3D imaging**.

## KEYWORDS

Seamless Monocular Metric Depth Estimation, Domain-aware Bin Estimation

**ACM Reference Format:**
Yihao Liu, Feng Xue, Anlong Ming, Mingshuai Zhao, Huadong Ma, and Nicu Sebe. 2024. SM$^4$Depth: Seamless Monocular Metric Depth Estimation across

Multiple Cameras and Scenes by One Model. In *Proceedings of Proceedings of the 32nd ACM International Conference on Multimedia (MM '24)*. ACM, New York, NY, USA, 10 pages. https://doi.org/10.1145/3664647.3681405

## 1 INTRODUCTION

Monocular depth estimation is a fundamental visual task with wide-ranging applications in the field of multimedia, such as video editing [7] image editing [17, 69, 71], 3D generation/synthesis [45, 47, 67], 3D reconstruction [55, 56], and human pose estimation [36, 50]. In this community, early research focused on MMDE [44, 49, 51, 58, 60, 70, 74], which were trained and tested only on specific datasets. However, they suffered from poor generalization when applied to unseen datasets, which limited their applications in the real world. To solve this issue, much research shifted their focus to monocular relative depth estimation (MRDE) [29, 38, 66] while disregarding the metric scale. Leveraging diverse and easily accessible relative depth data, these studies have achieved impressive performance, enabling their application in scale-free tasks, such as image editing [13, 34] and image stylization [22, 32], but not scale-sensitive applications, e.g. virtual reality [1, 33, 75], 3D reconstruction [52], and even robot navigation [8, 31, 61–63].

Beyond these approaches, universal MMDE has recently gained prominence for its generalization capabilities, marked by Depth Anything [64], ZoeDepth [5], and Metric3D [65]. However, these methods still face challenges in the two aspects of MMDE:

(1) **Inconsistent accuracy across scenes**: The real world varies widely in depth, ranging from $[1m, 2m]$ (close-up scenes) to $[0.5m, 80m]$ (street scenes), making models tend to focus on specific scenes and causing inconsistent accuracy across scenes.

(2) **Heavy reliance on data amount**: The reliance on massive training data (e.g. 8M metric depth data for Metric3D) remains due to the high complexity of determining a unique metric scale from a vast solution space of the natural scene.

Aiming to address these issues, we propose a Seamless Model for MMDE across Multiple cameras and scenes (SM$^4$Depth for short). First, based on explicit modeling of metric scale, we propose novel variation-based unnormalized depth bins which adaptively activate some parts rather than all of the fixed-length bins vector to describe the metric scale. This reduces the bin ambiguity inherent in previous width-based bins, and promotes the learning of widely different depth ranges in multiple scenes. Regarding the second issue, we propose a domain-aware bin estimation mechanism based on the "divide and conquer" idea, which estimates metric bins from various solution sub-spaces, not the entire one, for reducing complexity. **Divide**: we divide the common depth range into several range domains (RDs) offline and generate independent metric bins for each RD online. **Conquer**: we predict the RD that the input image belongs to and weightedly fuse all bins into a single one. To verify the accuracy consistency across diverse scenes, we propose BUPT Depth, a seamless RGB-D dataset, that consists of various indoor and outdoor scenes. Owing to the design of SM$^4$Depth, it performs superiorly on the consistency of accuracy across diverse scenes, which can be seen in Fig. 1. Notably, it also achieves comparative performance compared with the state-of-the-art zero-shot MMDE

methods but uses far fewer training samples (only 150K RGB-D pairs) and an affordable GPU (only single RTX 3090).

Our primary contributions are as follows:

(1) SM$^4$Depth achieves consistent accuracy across diverse scenes using a single model, eliminating the need for scene-specific parameters and pre-training. This enhances the practicality of MMDE in real-world applications.

(2) This paper tackles the long-term unresolved issue of bin ambiguity using variation-based depth bins. The proposed bins facilitate depth learning across scenes with significantly different depth ranges.

(3) Our domain-aware bin estimation mechanism reduces the reliance on massive training data in universal MMDE. This enables SM$^4$Depth to achieve state-of-the-art accuracy with 150K RGB-D training pairs (only **0.02%** amount used by previous methods) and a consumer-grade GPU.

(4) This paper presents the first no-clip RGB-D dataset. It is tailored to evaluate the consistent accuracy of MMDE methods **across diverse indoor and outdoor scenes**.

## 2 RELATED WORK

**Monocular metric depth estimation** is a classic visual task, in which determining metric scales is a crucial point, and there are two paradigms. Mainstream MMDE methods [11, 16, 24, 27, 28, 57, 72, 76] directly model this task as a pixel-wise regression problem (predicting continuous depth values in the real metric space), where metric scales are implicitly encoded. In contrast, since [19], several methods [3, 4, 46] have defined this task as a classification problem. Adabins [3] explicitly encoded the metric scale into image-level depth bins. We follow the latter paradigm as this paper focuses on recovering the metric scale. However, the same bin on two images with a large gap in depth range represents drastically different depths, causing misleading back-propagation during training. In this paper, we introduce variation-based bins to overcome this issue.
**Zero-shot generation** has become a new trend in monocular depth estimation in recent years. Early works [29, 39, 40] mainly achieved this goal by training with more accessible relative depth data. Initially, Li *et al.* [29] developed an MRDE pipeline on large-scale relative depth data. Ranftl *et al.* [40] trained an MRDE model on five datasets and re-applied the training strategy to [39]. For high practicality, the universal MMDE was first proposed in [5] which combined relative depth and metric depth to achieve generalization. Yin *et al.* [65] trained the model on 8M metric depth data for generalization. Yang *et al.* [64] proposed DepthAnything trained on 1M depth data and over 60M unlabeled data. Piccinelli *et al.* [37] designed a camera-adaptive model, UniDepth, and trained it on 300M metric depth images. This reliance on numerous training data is due to the complexity of determining correct metric scales from diverse scenes. Our approach aims to reduce this reliance.

## 3 PROBLEM ANALYSIS AND COUNTERMEASURES

In this section, we delve deep into the two issues of MMDE at the scene and data levels, and provide specific solutions for each issue.

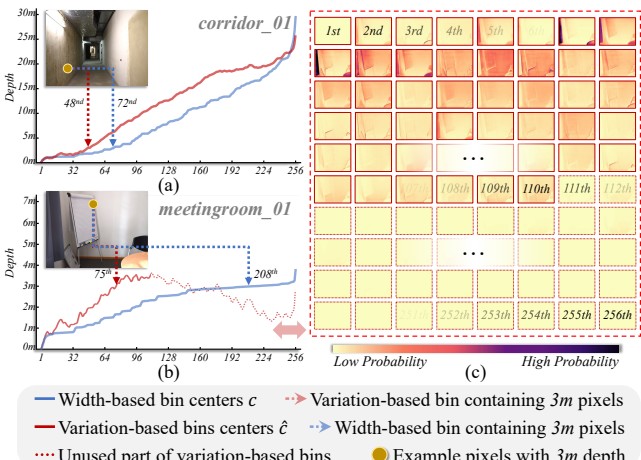

Figure 2: Two bin-center curves on (a) a distant-view image with range $[0m, 30m]$ and (b) a close-up one with range $[0m, 3.5m]$ from iBims-1 [23]. (c) represents probability maps $P$ corresponding to the red curve in (b).

## 3.1 Inconsistent Accuracy across Scenes

Generally, real-world images exhibit vastly different depth ranges, e.g. $[1m, 2m]$ for indoor close-up and $[0.5m, 80m]$ for street scenes. Such a large gap causes the model to overly concentrate on specific scenes instead of all scenes, leading to ***inconsistent accuracy across different scenes***. In this section, we solve this issue by novel variance-based depth bins that bridge the gap of metric scale representation across scenes. Before that, we briefly review the conventional depth bin [3] and outline its weakness.

**Reviewing width-based depth bin and its weakness.** Given the input image $I \in \mathbb{R}^{h \times w \times 3}$, Adabins [3] generates an $N$-channel probability map $P \in \mathbb{R}^{h \times w \times N}$ and a vector $c \in \mathbb{R}^{N \times 1}$ representing the centers of $N$ depth bins discreted from the depth interval, which are linearly combined to obtain a metric depth map $D \in \mathbb{R}^{h \times w}$:

$$D(i) = \sum_{n=1}^{N} c_n P_n(i) \tag{1}$$

where $D(i)$ is the $i$th pixel's predicted depth, and $P_n(i)$ denotes the probability for pixel $i$ that its depth is equal to the $n$th bin center $c_n$. In Eq. (1), the bin center $c_n$ is calculated by accumulating the width of each bin $b \in \mathbb{R}^{N \times 1}$:

$$c_n = d_{\min} + (d_{\max} - d_{\min}) \left( b_n/2 + \sum_{j=1}^{n-1} b_j \right) \tag{2}$$

where $b_n = (b'_n + \epsilon)/\sum_{i=1}^{N}(b'_i + \epsilon)$ denotes the normalized width of the $n$th depth bin, with $\epsilon = 10^{-3}$ and $b'_n \in [0, +\infty)$ being the unnormalized width predicted through a feedforward neural network (FFN) with the ReLU activation function. During training, the bi-directional Chamfer loss [18] is employed to enforce the small width $b'$ within the interesting depth interval in the ground truth depth map $\mathbf{D}$:

$$\mathcal{L}_{bin}(c, \mathbf{D}) = \sum_{\mathbf{d} \in \mathbf{D}} \min_{c_n \in c} ||\mathbf{d} - c_n||^2 + \sum_{c_n \in c} \min_{\mathbf{d} \in \mathbf{D}} ||\mathbf{d} - c_n||^2 \tag{3}$$

where $\mathbf{d}$ is the pixel's correct depth.

In natural scenes, the depth range of images varies significantly, yet all images must represent metric scales using the $N$ bins. This

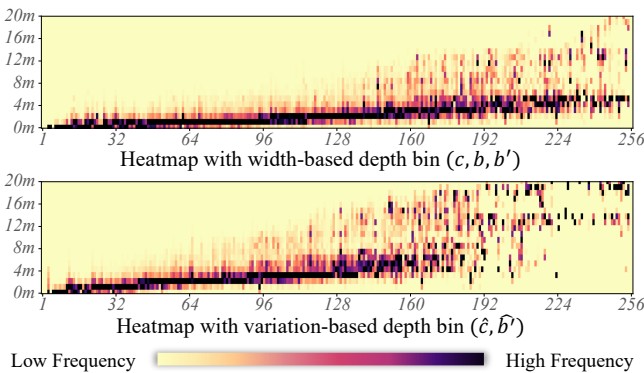

Figure 3: The heatmaps show the frequency of depth values occurring in each depth bin, which are obtained from iBims-1 [23]. If a square $(\mathcal{X}, \mathcal{Y})$ appears darker, it indicates that the depth value $\mathcal{Y}$ mainly occurs within the $\mathcal{X}^{\text{th}}$ depth bin.

leads to great variations in the metric depth represented by a single bin, a phenomenon we call "*bin ambiguity*". Taking Fig. 2 as an example, a 3m pixel would be classified in the 208th bin for an image with a range of $[0, 3.5m]$ (Fig. 2(b)), but in the $72^{th}$ bin for an image with a range of $[0, 30m]$ (Fig. 2(a)). Such an excessive gap would confuse the metric meaning of the probability map's channels and lead to the back-propagation of misleading signals during training.

**Depth variation-based bins for consistent accuracy.** The intuitive idea is to use the front portion of bins for short-range images and the entire bins for long-range images. To achieve this, we propose the variation-based unnormalized depth bins. Unlike the conventional bin $b'_n$, we use only an FFN without ReLU activation. In this way, the FFN outputs variations that allow negative values, denoted as $\hat{b}' \in \mathbb{R}^{N \times 1}$. Then, the bin center $c$ in Eq. (2) is re-formulated to an unnormalized bin center $\hat{c}$, which is no longer limited by the depth range of specific datasets (e.g., $[0m, 10m]$ for NYUDv2 and $[0m, 80m]$ for KITTI):

$$\hat{c}_n = \epsilon + \hat{b}'_n/2 + \sum_{j=1}^{n-1} \hat{b}'_j \tag{4}$$

**Analysis:** Since the depth variations $\hat{b}'$ are allowed to be negative, the bin center can reach the maximum depth on some intermediate bin $\hat{c}_n (\mathbf{n} < N)$ in short-range images, not necessarily the last bin center $\hat{c}_N$. Thus, all pixels can be fully expressed by the front bins $\{\hat{c}_n | n \in [1, \mathbf{n}]\}$, and do not have to involve the latter bins $\{\hat{c}_n | n \in (\mathbf{n}, N]\}$. As indicated by the red lines in Fig. 2, for the close-up image, the bin center reaches the maximum depth at the $110^{th}$ bin and then continues to decrease. While the later bins $\{\hat{c}_n | n \in (110, N]\}$ correspond to some probability channels close to zero $\{P_n | n \in (110, N]\}$, which can be observed through the visualization of part channels in Fig. 2 (c). This illustrates that these later bins are not actually used.

Fig. 3 presents additional statistics information, i.e., the frequency of depth values occurring in each depth bin. For the width-based depth bin $(c, b, b')$, depths below 4m occur most frequently across all bins. Conversely, variation-based depth bin $(\hat{c}, \hat{b}')$ exhibits larger depths in the latter bins. This means that the depth values represented by each bin center are pulled apart on the level of the entire dataset, suppressing the bin ambiguity.

## 3.2 Reliance on Massive Training Data

**Reason behind the reliance.** Practical applications differ from specific datasets in that images are taken from various camera angles and innumerable scenes. Due to the diverse nature of appearance, mapping from visual cues to a wide range of depth values becomes highly intricate and cannot be exhaustively presented. Consequently, determining metric depth bins entails exploring a vast solution space, which necessitates greater attention to reducing its complexity. However, previous works have overlooked this crucial aspect by directly making predictions (e.g., [5] solves for metric bins and [65] predicts metric depth) from the entire solution space, inevitably requiring massive training data. To address this issue, we first divide the whole solution space into several sub-spaces. Then, a "divide and conquer" method is proposed to generate metric bins in each sub-space and predict the best metric bins for the input.

**Stage 1: Online depth range domain generation.** To divide the solution space into sub-spaces, the previous approaches group all images according to semantic categories [5, 30]. However, a large gap in depth range may exist within one scene. Differently, we group all training images according to the depth range that better constrains the perspective and scene from which the image is taken.

According to [19], the amount of information for depth estimation decreases as the depth value increases. Thus, we employ a space-increasing strategy to gain more image groups (named range domain, RD) when the depth value is smaller. Assuming that the depth range is $[Z_{\min}, Z_{\max}]$ and there are $K$ RDs, the $k^{\text{th}}$ RD can be formulated as:

$$RD_k = \left[ Z_{\min}, Z_{\min} + \sum_{i=1}^{k} \frac{2i(Z_{\max} - Z_{\min})}{K(1+K)} \right] \tag{5}$$

We further visualize the RDs in the supplementary material *here*.

**Stage 2: Online domain-aware bin estimation design.** We design a domain-aware bin estimation mechanism that generates metric bins for each RD and finds the best-matching metric bins, following the "divide and conquer" idea in two steps.

**The "Divide" step** aims to discretize each depth interval $RD_k$ into $N$ bins. Specifically, given the deep feature of the input image, we leverage a transformer encoder to learn the relationship between the deep feature and $K$ preset learnable 1-D embeddings (called bin queries). The output embeddings of these queries are fed into an FFN to generate $K$ depth variation vectors $\{\hat{b}'^{[k]}|k\in[1,K]\}$, and calculate the bin center vectors $\{\hat{c}^{[k]}|k\in[1,K]\}$ using Eq. (4). To illustrate our idea, we compare three possible design choices:

- 1 **Query** + $K$ **FFNs**: Using $K$ FFNs to process the output of only one query.
- $K$ **Queries** + $K$ **FFNs**: Using $K$ FFNs to process the outputs of $K$ queries in a one-to-one way.
- $K$ **Queries** + 1 **FFN (Ours)**: Using only one FFN to process the outputs of $K$ queries.

The first two both employ $K$ FFNs. Thus, each FFN only learns the knowledge of a single RD during training, which leads to drastically different outputs of these FFNs and makes them sensitive to input noise. The last design is recommended as the best choice and the experiments (in Sec.6.4) verify its superiority over other options.

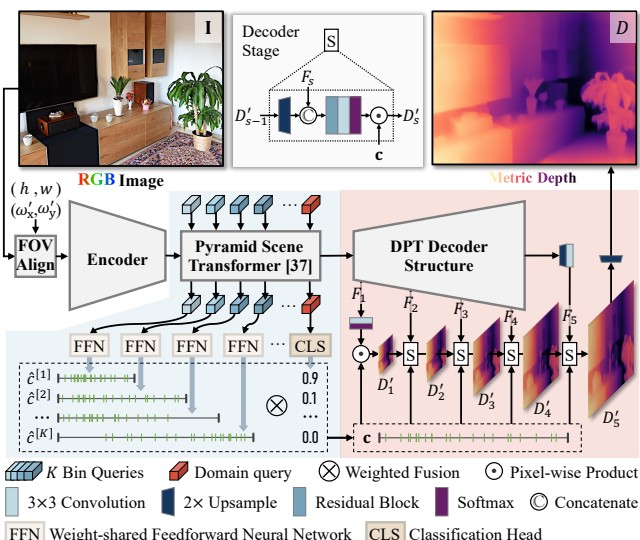

**Figure 4: SM$^4$Depth Pipeline containing the domain-aware bin estimation (blue mask) and the HSC-decoder (red mask).**

**The "Conquer" step** aims to estimate the correct RD for the input image and determine the best-matching metric bins. Specifically, we preset an additional 1-D embedding (called domain query) alongside the bin queries. Its corresponding output is then fed into a classification head (CLS) to generate the probability that the input image belongs to each RD, denoted as $\{y_k \in [0,1]|k \in [1,K]\}$.

Subsequently, considering the possibility of images being positioned near the decision boundary of RD classification, we do not select the top-scoring metric bin but instead combine all bin center vectors $\hat{c}^{[k]}$ to a single one by using the RD probabilities $\{y_k|k \in [1,K]\}$ as weights:

$$\mathbf{c} = \sum_{k\in[1,K]} \hat{c}^{[k]} y_k \tag{6}$$

where $\mathbf{c}$ is the final bin center vector.

## 4 ARCHITECTURE OF SM$^4$DEPTH

### 4.1 Pipeline.

Fig. 4 illustrates the structure of our network. Given an RGB image $\mathbf{I} \in \mathbb{R}^{\mathbf{h}\times\mathbf{w}\times 3}$, we first pre-process $\mathbf{I}$ to obtain a new image $I \in \mathbb{R}^{h\times w\times 3}$ with a unified field of view (FOV) (see Sec.4.2). Then, we extract the deep feature from $I$ by an encoder. Next, a pyramid scene transformer (PST) [46] is positioned between the encoder and decoder. It consists of three parallel transformer encoders with inputs of different patch sizes, respectively. We employ the transformer encoder with the smallest patch size to process all queries. Based on the mechanism in Sec.3.2, we obtain the bin center vector $\mathbf{c}$ of image $I$. Finally, we design a decoder with hierarchical scale constraints (HSC-Decoder) to anchor the metric scale in multiple resolutions and output the metric depth map $D$ (see Sec.4.3).

### 4.2 FOV alignment Pre-Processing.

According to [65], eliminating "metric ambiguity" is the key to achieving universal MMDE. Therefore, we pre-process all input images to align their field of view (FOV). Given an input image

**Figure 5: Top view of the scene where we collected the BUPT Depth dataset. Red lines indicate indoor scenes and purple lines indicate outdoor scenes. We give five images and their ground truth depth calculated by CREStereo [26] as examples.**

**I** with focal length $(\mathbf{f_x}, \mathbf{f_y})$, we first preset the input resolution of network as $(h, w)$ and define the target FOV as $(\omega'_\mathbf{x}, \omega'_\mathbf{y})$ in radians.

Then, a rectangular region $I' \in \mathbb{R}^{h' \times w' \times 3}$ on **I** equivalent to the target FOV $(\omega'_\mathbf{x}, \omega'_\mathbf{y})$ is calculated by the FOV equation:

$$w' = 2\mathbf{f_x}\tan(\omega'_\mathbf{x}/2) \; , \; h' = 2\mathbf{f_y}\tan(\omega'_\mathbf{y}/2) \tag{7}$$

Next, we crop this region $I'$ from the original image **I**, and fill the pixels beyond **I** with 255. Finally, the region $I'$ is resized to the target resolution $(h, w)$ to generate a new image $I$ as input of the network. Note that this FOV alignment is similar to the CSTM_image [65] in result, but without maintaining a canonical camera space. Thus it is more straightforward.

## 4.3 Decoder with hierarchical scale constraints.

Our decoder draws inspiration from the refinement decoder structures [9, 25], but the divergence lies in scale constraints on the metric depth at each stage. As shown in Fig. 4, taking the PST's output, denoted as $F_1$, as input, we employ the DPT's decoder [39] to gradually recover the resolution of features, denoted as $F_s$ with a size $\frac{h}{2^{(6-s)}} \times \frac{w}{2^{(6-s)}}$, where $s \in \{1, 2, 3, 4, 5\}$ is the stage number. In the first stage, $F_1$ is compressed into $N$-channel and then multiplied pixel-wisely with **c** by Eq. (1), generating a low-resolution depth map $D'_1 \in \mathbb{R}^{\frac{h}{32} \times \frac{w}{32}}$. In the following $s^{\text{th}}$ stage, the depth map of the former stage $D'_{s-1}$ is upsampled and fused with feature $F_s$ by a residual convolution block [9]. Then we linearly combine the fused feature and the bin centers **c** to generate the depth map $D'_s \in \mathbb{R}^{\frac{h}{2^{(6-s)}} \times \frac{w}{2^{(6-s)}}}$. In this way, the depth map of the last stage $D'_5 \in \mathbb{R}^{\frac{h}{2} \times \frac{w}{2}}$ is obtained. Compared to the previous refinement decoder [9], the HSC-Decoder incorporates the metric bins into each stage to progressively refine the depth range, thus performing better in recovering the depth range. The loss functions are further described in the supplementary material *here*.

## 5 UNCUT RGBD DATASET: BUPT DEPTH

BUPT Depth (see in Fig. 5) is proposed to evaluate consistency in accuracy across indoor and outdoor scenes, including streets, canteen, classroom, and lounges, etc. This dataset shows a variety of lighting, scenes, and viewing angles, making depth estimation challenging. It consists of 14,932 continuous RGB-D frames captured in BUPT by ZED2. In addition to the outputs of ZED2, we provide the re-generated depth maps from CREStereo [26] and the sky segmentation from ViT-Adapter [10]. The color and depth streams are captured with intrinsics of 1091.517 and a baseline of 120.034mm.

## 6 EXPERIMENTS

### 6.1 Experimental setting

**Datasets:** For training, we randomly sample RGB-Depth pairs from various datasets. Specifically, we sample 24K pairs from ScanNet [15], 15K pairs from Hypersim [41], 51K pairs from DIML [12], 36K pairs from UASOL [2], 14K pairs from ApolloScape [21], and 11K pairs from CityScapes [14]. During training, NYUD [35] and KITTI [53] are used for validation. In addition, we apply the same pre-processing steps to the training data as [5, 40], elaborated in the supplementary material *here*. During testing, we employ eight datasets not seen during training: SUN RGB-D [48], iBims-1 [23], ETH3D Indoor/Outdoor [43], DIODE Indoor/Outdoor [54], nuScenes-val [6], and DDAD [20]. Note that we remove the test set of NYUD from SUN RGB-D for a fair comparison.

**Metrics:** We employ four metrics [3] for evaluation: the accuracy under threshold ($\delta_k < 1.25^k, k = 1, 2, 3$), the absolute relative error (REL), and the root mean squared error (RMSE). In addition, we use the relative improvement across datasets (mRI$_\eta$) and metrics (mRI$_\theta$) in [5]. During the evaluation, the final output is obtained by averaging the predictions for an image and its mirror image. In addition, the final output is upsampled to match the original image size, and all metrics are computed within the same FOV.

**Implementation Details:** SM$^4$Depth employs the Swin Transformer Base as the backbone, and runs on a single NVIDIA RTX 3090 GPU. The network is trained by the Adam optimizer with parameters $(\beta_1, \beta_2) = (0.9, 0.999)$. The training runs for 20 epochs with a batch size of 10. The initial learning rate is set to $2 \times 10^{-5}$ and gradually reduced to $2 \times 10^{-6}$. Note that, an over-large fixed FOV would cause too large invalid area in the small FOV dataset, making the network underfitting. We empirically set the fixed FOV to $(\omega'_\mathbf{x}, \omega'_\mathbf{y}) = (58°, 45°)$ and the fixed resolution to $(w, h) = (564, 424)$.

### 6.2 Result on BUPT Depth

Table 1 presents a quantitative comparison between SM$^4$Depth and state-of-the-art methods, categorized based on different ground truth. SM$^4$Depth achieves superior performance in most metrics when utilizing ground truth from either ZED2 or CREStereo [26]. Notably, SM$^4$Depth is trained on a smaller dataset of 0.015M pairs (only 0.02%~5% of the data used by previous methods), with FLOPs and parameter count typically half that of prior methods.

Fig. 6 provides a more intuitive comparison between SM$^4$Depth and state-of-the-art methods from a time-series perspective, visualizing the RMSE of each frame using two line graphs. SM$^4$Depth (orange) achieves a relatively high accuracy regardless of indoor

| Method | Backbone | Training Pairs | FLOPs | Params | BUPT Depth - Ground Truth gained by ZED2 | | | | | | BUPT Depth - Ground Truth gained by CREStereo | | | | | |
|---|---|---|---|---|---|---|---|---|---|---|---|---|---|---|---|---|
| | | | | | $\delta_1 \uparrow$ | $\delta_2 \uparrow$ | $\delta_3 \uparrow$ | REL$\downarrow$ | RMSE$\downarrow$ | log10$\downarrow$ | $\delta_1 \uparrow$ | $\delta_2 \uparrow$ | $\delta_3 \uparrow$ | REL$\downarrow$ | RMSE$\downarrow$ | log10$\downarrow$ |
| ZoeD-NK [5] | BEiT-L | - | 283G | 267M | 0.314 | 0.485 | 0.576 | 1.006 | 9.758 | 0.263 | 0.365 | 0.505 | 0.579 | 0.974 | 9.377 | 0.252 |
| Metric3D [65] | CNXT-L | over 8M | 569G | 203M | 0.155 | 0.318 | 0.417 | 2.446 | 23.794 | 0.427 | 0.293 | 0.402 | 0.470 | 1.816 | 22.736 | 0.352 |
| DepthAnything-NK [64] | ViT-L | over 61M | 451.9G | 335.3M | 0.193 | 0.3623 | 1.000 | 1.1362 | 7.9838 | 0.2978 | 0.221 | 0.3742 | 1.000 | 1.120 | 7.833 | 0.292 |
| UniDepth [37] | ViT-L | roughly 3M | - | 347M | 0.311 | 0.598 | 1.000 | 0.6559 | 8.399 | 0.1933 | 0.109 | 0.396 | 1.000 | 0.983 | 9.422 | 0.264 |
| **SM$^4$Depth** | **Swin-B** | **0.015M** | **105G** | **110M** | **0.536** | **0.805** | 0.924 | **0.295** | **3.440** | **0.118** | **0.629** | **0.875** | 0.966 | **0.241** | **2.888** | **0.094** |

**Table 1: Quantitative results on BUPT Depth. Comparisons are conducted with the ground truths of ZED and CREStereo, respectively. The best results are in bold, and the second-best ones are underlined. - indicates unknown numbers.**

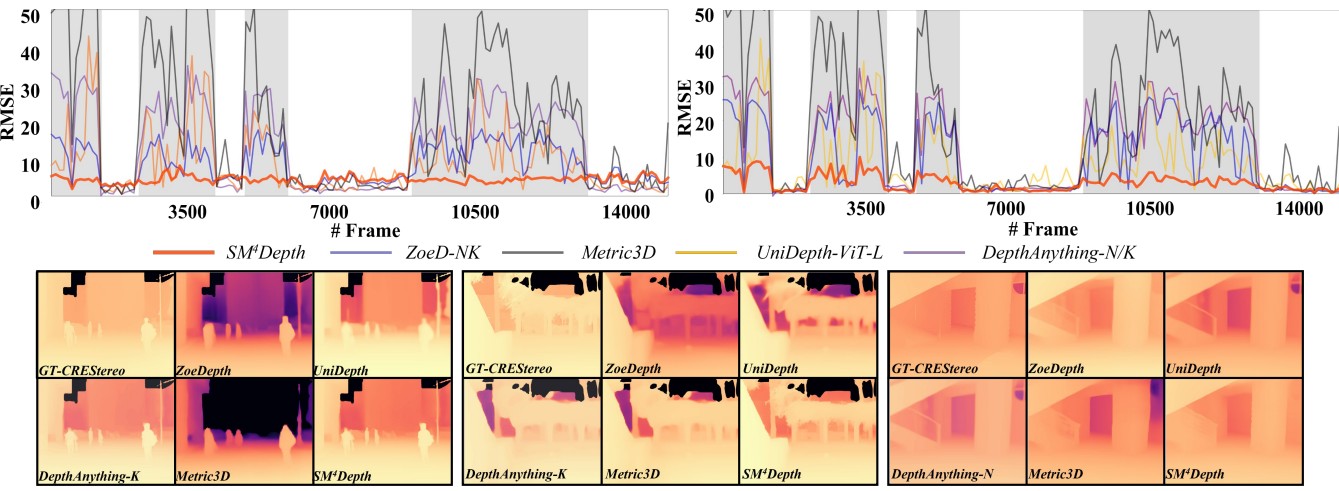

**Figure 6: RMSE per frame of SM$^4$Depth (orange), Metirc3D (gray), ZoeDepth-NK (blue), UniDepth (yellow), DepthAnything-NK(purple) on BUPT Depth. We use the stereo depth of ZED2 (the first chart) and CREStereo[26] (the second chart) as ground truth, respectively. Gray indicates outdoor frames, and white indicates indoor frames.**

or outdoor scenes on both two ground truth. In contrast, other methods fail to achieve high accuracy simultaneously indoors and outdoors. Specifically, ZoeD-NK, Metric3D, and UniDepth show fluctuating accuracy in outdoor scenes, with some even exceeding over 50 at times. And the accuracy of SM$^4$Depth varies more smoothly. In the first two examples we presented in Fig. 6, both SM$^4$Depth and UniDepth obtain a more accurate metric scale than others, but SM$^4$Depth provides sharper outputs, even enabling the wooden fence in frame 9957 to be discernible. In the indoor scene, SM$^4$Depth produces clearer outlines than other methods.

## 6.3 Comparison to the state of the art

*6.3.1 Quantitative Result.* We employ two classical MMDE methods, i.e., BTS [25] and AdaBins [3], as well as two more advanced MMDE approaches, i.e., NeWCRFs [68] and MIM [73], for comparison. Moreover, we also employ several universal MMDE methods, i.e., Metric3D [65], ZoeDepth [5], DepthAnything [64], and UniDepth [37], for comparison (N indicates NYUD fine-tuning and K for KITTI fine-tuning; they are also applicable to SM$^4$Depth).

In Table 2, the upper part shows the zero-shot performance on four indoor datasets, and the lower part shows that on four outdoor datasets. SM$^4$Depth not only achieves accuracy consistency across indoor and outdoor but also outperforms most MMDE methods on public datasets and competes with top-performing algorithms like UniDepth and DepthAnything. Compared to the earliest universal MMDE algorithm, ZoeDepth, SM$^4$Depth demonstrates superior accuracy across all datasets, leading in both absolute metrics

($\delta_1$, RMSE) and relative metrics (REL). This indicates SM$^4$Depth's ability to learn more accurate relative depth from metric depth data. Compared to Metric3D, SM$^4$Depth performs better on most datasets, (i.e., SUN RGB-D, ETH3D, DIODE, and DDAD) and similar on iBims-1, but is only trained 150K images, which proves the effectiveness of SM$^4$Depth. Especially, SM$^4$Depth outperforms Metric3D by +58.08% and +8.10% mRI$_\theta$ on ETH3D Outdoor and DDAD. In addition, SM$^4$Depth outperforms Metric3D on nuScenes-val by -1.285 of RMSE, but falls behind on $\delta_1$ and REL, as Metric3D is trained on much more self-driving datasets, which endows it an advantage in such scenes. Notably, SM$^4$Depth outperforms Metric3D by +58.08% and +8.10% mRI$_\theta$ on ETH3D Outdoor and DDAD, respectively. Although not surpassing UniDepth in accuracy overall, SM$^4$Depth closely approaches or even exceeds it on some datasets (iBims-1, ETH3D Indoor/Outdoor).

Table 3 displays the results on NYUD and KITTI. With the zero-shot setting, our method obtains lower $\delta_1$ and higher RMSE than Metric3D on NYUD and KITTI. However, after being fine-tuned on NYUD and KITTI, SM$^4$Depth achieves competitive accuracy with the state-of-the-art methods, while avoiding a significant degradation in accuracy on zero-shot datasets (see in Table 2).

*6.3.2 Qualitative Result.* Fig. 7 visualizes several methods' predictions and depth distributions. The $1^{st} - 3^{rd}$ columns show close-up scenes challenging depth range determination. Previous methods obtain incorrect depth distributions, while Metric3D tends to push the background farther when the foreground boundary is clearly

| Categories | Method | SUN RGB-D | | | | iBims-1 Benchmark | | | | ETH3D Indoor | | | | DIODE Indoor | | | |
|---|---|---|---|---|---|---|---|---|---|---|---|---|---|---|---|---|---|
| | | $\delta_1 \uparrow$ | REL $\downarrow$ | RMSE $\downarrow$ | mRI$_\theta \uparrow$ | $\delta_1 \uparrow$ | REL $\downarrow$ | RMSE $\downarrow$ | mRI$_\theta \uparrow$ | $\delta_1 \uparrow$ | REL $\downarrow$ | RMSE $\downarrow$ | mRI$_\theta \uparrow$ | $\delta_1 \uparrow$ | REL $\downarrow$ | RMSE $\downarrow$ | mRI$_\theta \uparrow$ |
| **Full-shot** | BTS [25] | 0.718 | 0.181 | 0.533 | -31.45% | 0.536 | 0.233 | 1.059 | -32.82% | 0.360 | 0.324 | 2.210 | -18.73% | 0.208 | 0.419 | 2.382 | -34.57% |
| | AdaBins [3] | 0.751 | 0.167 | 0.493 | -23.07% | 0.548 | 0.216 | 1.078 | -29.56% | 0.283 | 0.361 | 2.347 | -31.23% | 0.173 | 0.442 | 2.450 | -40.95% |
| | NeWCRFs [68] | 0.779 | 0.159 | 0.437 | -14.67% | 0.543 | 0.209 | 1.031 | -26.43% | 0.452 | 0.268 | 1.874 | 0.76% | 0.183 | 0.402 | 2.307 | -33.51% |
| | MIM [59] | 0.844 | 0.147 | 0.341 | 0.00% | 0.717 | 0.163 | 0.813 | 0.00% | 0.453 | 0.287 | 1.800 | 0.00% | 0.416 | 0.317 | 1.960 | 0.00% |
| **Zero-shot** | ZoeD-N [5] | 0.850 | 0.125 | 0.357 | 3.66% | 0.652 | 0.171 | 0.883 | -7.53% | 0.388 | 0.275 | 1.678 | -1.13% | 0.376 | 0.331 | 2.198 | -8.72% |
| | DepthAnything-N [64] | 0.897 | 0.107 | 0.272 | 17.90% | 0.716 | 0.150 | 0.726 | 6.17 % | 0.438 | 0.250 | 1.663 | 5.73 | 0.277 | 0.339 | 2.113 | -16.05 |
| | **SM[4]Depth-N** | 0.874 | 0.121 | 0.303 | 10.80% | 0.715 | 0.162 | 0.801 | 0.60% | 0.486 | 0.249 | 1.662 | 9.40% | 0.418 | 0.298 | 1.790 | 5.05% |
| | ZoeD-NK [5] | 0.841 | 0.129 | 0.367 | 1.42% | 0.610 | 0.189 | 0.952 | -15.99% | 0.353 | 0.280 | 1.691 | -4.53% | 0.386 | 0.335 | 2.211 | -8.57% |
| | Metric3D[65] | 0.033 | 2.631 | 5.633 | × | **0.818** | 0.158 | **0.582** | **15.18%** | **0.536** | 0.335 | 1.550 | 5.16% | 0.505 | 0.427 | 1.687 | 0.21% |
| | UniDepth-ViT-L [37] | **0.953** | **0.089** | **0.232** | **28.11%** | 0.262 | 0.344 | 1.104 | -70.09% | 0.177 | 0.503 | 2.126 | -51.43 | **0.762** | **0.186** | **1.290** | **52.89%** |
| | **SM[4]Depth** | 0.869 | 0.127 | 0.301 | 9.43% | 0.790 | **0.134** | 1.407 | 15.06% | 0.527 | 0.233 | 1.407 | 18.99% | 0.356 | 0.300 | 1.721 | 1.04% |

| Categories | Method | nuScenes-val | | | | DDAD | | | | ETH3D Outdoor | | | | DIODE Outdoor | | | |
|---|---|---|---|---|---|---|---|---|---|---|---|---|---|---|---|---|---|
| | | $\delta_1 \uparrow$ | REL $\downarrow$ | RMSE $\downarrow$ | mRI$_\theta \uparrow$ | $\delta_1 \uparrow$ | REL $\downarrow$ | RMSE $\downarrow$ | mRI$_\theta \uparrow$ | $\delta_1 \uparrow$ | REL $\downarrow$ | RMSE $\downarrow$ | mRI$_\theta \uparrow$ | $\delta_1 \uparrow$ | REL $\downarrow$ | RMSE $\downarrow$ | mRI$_\theta \uparrow$ |
| **Full-shot** | BTS [25] | 0.420 | 0.285 | 9.140 | -9.24% | 0.802 | 0.146 | 7.611 | -13.07% | 0.175 | 0.831 | 5.746 | 7.19% | 0.172 | 0.838 | 10.475 | -34.70% |
| | AdaBins [3] | 0.483 | 0.272 | 10.178 | -7.45% | 0.757 | 0.155 | 8.673 | -22.80% | 0.110 | 0.889 | 6.480 | -12.65% | 0.162 | 0.853 | 10.322 | -36.09% |
| | NeWCRFs [68] | 0.415 | 0.280 | 7.402 | -0.64% | 0.866 | 0.120 | 6.359 | 2.66% | 0.258 | 0.799 | 5.061 | 29.57% | 0.177 | 0.841 | 9.304 | -29.25% |
| | MIM [59] | 0.396 | 0.283 | 6.868 | 0.00% | 0.859 | 0.134 | 6.157 | 0.00% | 0.159 | 0.889 | 6.048 | 0.00% | 0.269 | 0.625 | 7.819 | 0.00% |
| **Zero-shot** | ZoeD-K [5] | 0.379 | 0.290 | 6.900 | -2.41% | 0.833 | 0.130 | 7.154 | -5.41% | 0.303 | 1.012 | 5.853 | 26.65% | 0.269 | 0.823 | 6.891 | -6.60% |
| | DepthAnything-K [64] | 0.579 | 0.223 | 5.844 | 27.44% | 0.840 | 0.118 | 6.953 | -1.06% | 0.193 | 0.897 | 6.423 | 4.76% | 0.309 | 0.836 | 7.599 | -5.35% |
| | **SM[4]Depth-K** | 0.623 | 0.229 | 7.175 | 23.98% | 0.841 | 0.160 | 5.677 | -4.56% | **0.452** | 0.294 | **3.168** | **99.61%** | **0.280** | 0.552 | 8.335 | 3.06% |
| | ZoeD-NK [5] | 0.371 | 0.299 | 6.988 | -4.57% | 0.821 | 0.139 | 7.274 | -8.77% | 0.337 | 0.752 | 4.758 | 49.56% | 0.207 | 0.735 | 7.570 | -12.49% |
| | Metric3D*[65] | 0.868 | 0.143 | 8.506 | 48.27% | 0.896 | 0.119 | 7.262 | -0.01% | 0.324 | 0.724 | 9.830 | 19.93% | 0.169 | 0.499 | 9.353 | -12.21% |
| | UniDepth-ViT-L [37] | **0.921** | **0.088** | **4.270** | **79.76%** | **0.935** | **0.103** | 5.062 | **16.58%** | 0.424 | 0.341 | 4.060 | 87.05% | **0.597** | **0.483** | 5.631 | **57.54%** |
| | **SM[4]Depth** | 0.672 | 0.214 | 7.221 | 29.65% | 0.890 | 0.123 | **5.390** | 8.09% | 0.348 | **0.273** | 3.274 | 78.01% | 0.190 | 0.487 | 8.435 | -5.05% |

**Table 2: Quantitative results on zero-shot datasets. mRI$_\theta$ denotes the mean relative improvement compared to MIM across all metrics($\delta_1$, REL, RMSE). All methods undergo evaluation consistently within a specific region. The best results are in bold and the second-best ones are underlined. × indicates poor performance. * means that Metric3D was trained on DDAD.**

| Categories | Method | Backbone | NYUD | | | | | | | KITTI | | | | | | |
|---|---|---|---|---|---|---|---|---|---|---|---|---|---|---|---|---|
| | | | $\delta_1 \uparrow$ | $\delta_2 \uparrow$ | $\delta_3 \uparrow$ | REL$\downarrow$ | RMSE$\downarrow$ | log10$\downarrow$ | mRI$_\theta$ | $\delta_1 \uparrow$ | $\delta_2 \uparrow$ | $\delta_3 \uparrow$ | REL$\downarrow$ | RMSE$\downarrow$ | log10$\downarrow$ | mRI$_\theta$ |
| **Full-shot** | ZoeD-N/K [5] | BEiT-L | 0.956 | 0.995 | 0.999 | 0.075 | 0.279 | 0.032 | 0.00% | **0.978** | **0.998** | 0.999 | **0.049** | **2.221** | **0.021** | **0.00%** |
| | ZoeD-NK [5] | BEiT-L | 0.954 | 0.996 | 0.999 | 0.076 | 0.286 | 0.033 | -1.18% | 0.971 | 0.994 | 0.996 | 0.053 | 2.415 | 0.024 | -5.43% |
| | DepthAnything-N/K [64] | ViT-L | 0.983 | **0.998** | **1.000** | 0.055 | 0.212 | 0.024 | 13.15% | 0.975 | 0.996 | **1.000** | 0.057 | 2.443 | 0.024 | -6.83% |
| | **SM[4]Depth-N/K** | Swin-B | 0.932 | 0.991 | 0.998 | 0.088 | 0.328 | 0.038 | -9.44% | 0.971 | 0.996 | 0.999 | 0.054 | 2.477 | 0.023 | -5.36% |
| **Zero-shot** | Metric3D [65] | CNXT-L | 0.926 | 0.984 | 0.995 | 0.091 | 0.340 | 0.038 | -11.09% | 0.962 | 0.993 | 0.998 | 0.060 | 2.969 | 0.026 | -13.69% |
| | UniDepth | ViT-L | **0.984** | 0.997 | **1.000** | **0.053** | **0.208** | **0.023** | **14.35%** | 0.975 | 0.996 | **1.000** | **0.049** | 2.476 | **0.021** | -1.98% |
| | **SM[4]Depth** | Swin-B | 0.860 | 0.981 | 0.997 | 0.126 | 0.417 | 0.052 | -31.93% | 0.928 | 0.985 | 0.996 | 0.087 | 3.272 | 0.038 | -35.42% |

**Table 3: Quantitative result on NYUD and KITTI. All methods undergo evaluation in a consistent region. The best results are in bold and the second-best ones are underlined.**

delineated. The 4[th] and 5[th] columns show indoor scenes containing a large area of wall. Other methods suffer from incorrect depth range, while SM[4]Depth recovers the depth distribution accurately. The 6[th] and 7[th] columns show two close-up outdoor scenes. The predictions of ZoeDepth and DepthAnything exhibit overall shifts, while Metric3D fails to distinctly differentiate between the front objects and the wall. Due to training on multiple metric depth datasets, SM[4]Depth generates a visually reasonable depth distribution while it does not assign an extreme depth value to sky regions because they are set to 0 during training. The last two columns show images from self-driving scenes. Although all methods generate good depth maps, SM[4]Depth obtains a more accurate depth distribution and captures richer details than other methods. Especially in the 9[th] column, where objects are up to 80m away, our method correctly predicts their farthest depths as well as generating fine tree trunk edges.

## 6.4 Detail Analysis

*6.4.1 Number of depth range domain.* We explore the optimal number of RD, i.e., $K$, and additionally evaluate the uniform partition

| V-Bin[1] | WF-Bin[2.1] | DBE[2.2] | HSC[3] | iBims-1 | ETH3D | DIODE | DDAD | mRI$_\eta \uparrow$ |
|---|---|---|---|---|---|---|---|---|
| √ | √ | √ | √ | **0.673** | 2.373 | 5.605 | 5.390 | **10.92%** |
| × | √ | √ | √ | 0.692 | 2.504 | 6.033 | 5.726 | 6.03% |
| × | × | √ | √ | 0.701 | 2.692 | 6.111 | 5.486 | 4.53% |
| × | × | × | √ | 0.741 | 2.566 | 6.163 | 5.587 | 3.67% |
| × | × | × | × | 0.695 | 2.695 | 6.107 | 6.767 | 0.00% |

1: *Depth-Variation based Bin*  2.2: *Domain-aware Bin Estimation*
2.1: *Weighted Fusion of Bins*  3: *Decoder with Hierarchical Scale Constraints*

**Table 4: RMSE results of the ablation study. The best results are in bold, while the second-best ones are underlined.**

strategy [19] when using the best $K$. Fig. 8 shows all variants' performance on the mixing test sets. As $K$ increases, RMSE decreases slowly. RMSE suddenly drops below 3.4 when $K = 4$ and increases again at 5 or 6. We argue that this phenomenon occurs because RDs better describe images with different appearance when $K = 4$ and prevent excessive similarity between RDs due to redundant division. In addition, using the uniform partition strategy leads to a notable decrease in $\delta_1$ and RMSE.

*6.4.2 Ablation study.* We conduct the ablation study by gradually removing our designs and comparing all variants on the mixing test sets. In Table 4, the baseline (last row) consists of only an

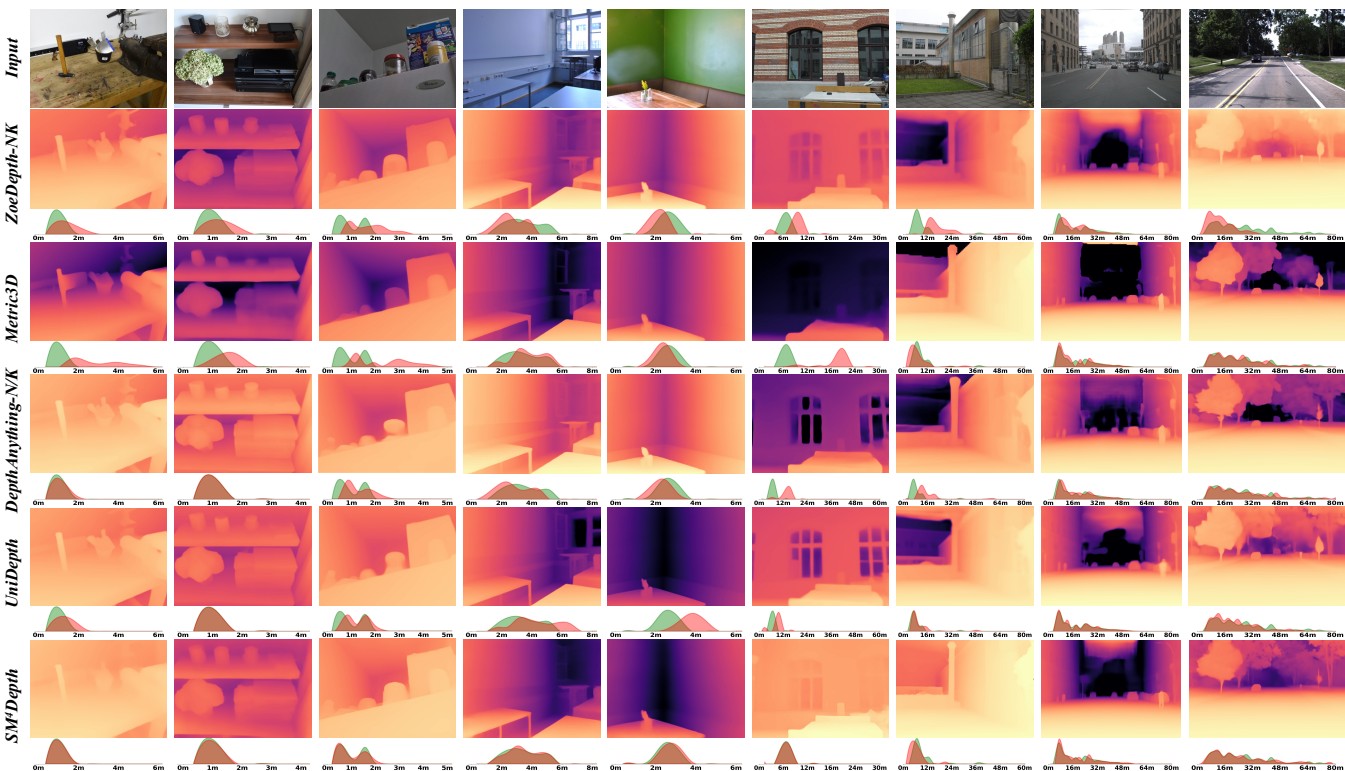

**Figure 7: Qualitative comparison with MDE methods on zero-shot datasets. The depth distribution is under the depth maps with green for ground truth and red for prediction.**

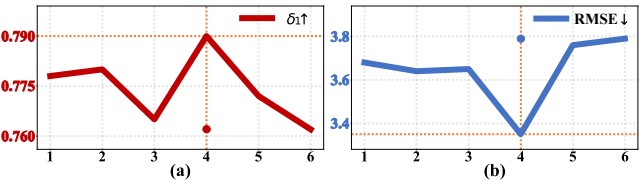

(a)

(b)

**Figure 8: Parameter experiment about $K$. The performance, as measured by $\delta_1$ and RMSE, is optimal when $K$ equals 4. The dots represent the use of the uniform partition strategy.**

| Design Choices | iBims-1 | ETH3D | DIODE | DDAD | mRI$_\eta$ ↑ |
|---|---|---|---|---|---|
| 1 * Query + $K$ * FFNs | 0.770 | 2.522 | 5.982 | 6.601 | 0.00% |
| $K$ * Queries + $K$ * FFNs | 0.734 | 2.401 | 5.820 | 6.920 | 1.84% |
| $K$ * Queries + 1 * FFN | **0.673** | **2.373** | **5.605** | **5.390** | **10.79%** |

**Table 5: RMSE results of DBE. The best results are in bold, while the second-best ones are underlined.**

encoder-decoder structure and a Pyramid Scene Transformer [46]. Observably, the RMSEs increase overall as the proposed modules and innovations are gradually removed. The depth-variation based bins make the greatest contribution (+4.89% mRI$_\eta$), indicating its effectiveness in learning large depth range gaps. The entire domain-aware bin estimation increases mRI$_\eta$ by 2.36%, with the weighted fusion scheme contributing 1.5% of this. In addition, the HSC-decoder improves mRI$_\eta$ by 3.67%.

*6.4.3 Comparing designs for domain-aware bin estimation.* As shown in Table 5, we compare three design choices of our domain-aware bin estimation mentioned in Sec.3.2 on the same four datasets in the ablation study. Compared to the other settings, "$K$*Query+1*FFN" achieves the lowest RMSE and highest mRI$_\eta$, and outperforms other variants by a large margin. The reason is that the single FFN

is trained on multiple RDs and thus learns common knowledge for bin estimation from multiple RDs.

## 7 CONCLUSION

This paper proposes a seamless MMDE algorithm, SM$^4$Depth, to solve the problems of inconsistent accuracy across diverse scenes and reliance on massive training data. Firstly, we discuss the inherent issue of the bin-based methods when learning depth range with large gap, that is the large inconsistency of the same bin in different images. To address this issue, we propose the variation-based depth bins that allow the network to effectively learn scenes with different depth ranges. Next, to reduce the complexity of estimating correct metric bins from a vast solution space, this paper designs a "divide and conquer" method to determine metric bins from multiple solution sub-spaces, thereby reducing the network's reliance on massive training data. Finally, we propose an uncut depth dataset, BUPT Depth, to verify the accuracy consistency across scenes. Our method obtains outstanding performance with only 150K RGB-D pairs for training and achieves accuracy consistency.

## 8 ACKNOWLEDGMENT

This work was supported by the Innovation Research Group Project of NSFC (61921003), the MUR PNRR project FAIR (PE00000013) funded by the NextGenerationEU and by the PRIN project CREATIVE (Prot.2020ZSL9F9).

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
