# OpenReview forum: "SM4Depth: Seamless Monocular Metric Depth Estimation across Multiple Cameras and Scenes by One Mode"
_acmmm.org/ACMMM/2024/Conference — MM2024 Poster_

### Official Review · Reviewer_8gm1 · 2024-04-30

**Rating:** 4
**Confidence:** 4

**Summary:**

This paper has proposed a novel metric scale modeling method for universal depth estimation with a single model. A new dataset is collected by using the ZED stereo camera. Extensive experiments are conducted which verify that the proposed SM4Depth model achieves competitive accuracy and generalization capacity on unseen data, while trained on 150K RGB-D image pairs. The source code has also been made available to foster future research.

**Strengths:**

1. A large-scale dataset is collected using the ZED stereo dataset.
2. The paper is overall well written and presents extensive experimental analyses, which verifies the effectiveness of the proposed method for universal depth estimation using a single camera.
3. The generalization capacity of the proposed model is verified in diverse scenes.
4. The proposed method succeeds in achieving state-of-the-art accuracy using only 150K RGB-D training pairs.

**Limitations:**

1. While the generalization capacity in diverse scenes has been verified, it would be nice to show some results in adverse scenes for analysis, which is an important aspect for assessing generalizability in real-world applications. For example, results in adverse weather conditions and specular surfaces could be discussed.
2. As the proposed dataset leverages the ZED camera for data collection, how to ensure that the depth estimation ground truth is reliable?
3. In Table 1, the Depth Anything model could be compared for analysis.
4. Some failure cases and challenging cases of such universal depth estimation models could be presented.
5. While this work follows a setting of a fixed FoV, it would be nice to discuss whether the method can scale to other cameras like active stereo cameras and panoramic cameras with wide FoV.
6. In the main paper, the description of the dataset should be greatly enriched as this is one of the important contributions of this paper. The distribution of different scenes and the diversity of the conditions covered in the dataset could be presented.

**Suitability:**

3

---

### Official Review · Reviewer_Qwxg · 2024-05-19

**Rating:** 4
**Confidence:** 3

**Summary:**

This paper introduces a novel metric scale modeling method that reduces the graph channel ambiguity caused by width-based depth bins under large depth gaps in training data. Additionally, a divide-and-conquer strategy is employed to achieve training on consumer-grade GPUs. Furthermore, the authors propose a new dataset.

**Strengths:**

1) This method appears to be highly lightweight, with significant reductions in parameter count and computational cost compared to other approaches.

2) The use of variation-based depth bins to address bin ambiguity issues sounds very interesting.

**Limitations:**

1) The authors should provide more detailed explanations in the paper regarding the differences between this dataset and other existing datasets.

2) Why is there such a significant different performance between SM4Depth and Metric3D/Unidepth on Campus Depth and other datasets?

3) The Depth variation based bins proposed in the paper may have certain limitations. It appears that they may not be as effective for short-range depth estimation. It would be beneficial if the authors could provide additional evidence of its effectiveness from various perspectives.

**Suitability:**

3

---

### Official Review · Reviewer_csKn · 2024-05-24

**Rating:** 4
**Confidence:** 3

**Summary:**

This paper introduces SM4Depth, a model for consistent monocular depth estimation in various environments. It uses variation-based depth bins and domain-aware bin estimation to improve accuracy and reduce the need for extensive training data. The model's effectiveness is demonstrated by the experiments across multiple datasets. Additionally, the authors introduce a new dataset, Campus Depth, for robust evaluation of depth estimation models.

**Strengths:**

S1：SM4Depth achieves high performance with significantly fewer training data and only requires a single consumer-grade GPU for training.

S2：The well-motivated focus on the issue of depth consistency enhances the model's reliability and applicability.

S3：Extensive experimentation with various results supports the effectiveness of the proposed method.

**Limitations:**

W1: The authors claim their method has advantages in terms of training sample size, GPU usage, computational complexity, and parameter count. However, the experimental section seems to lack comparisons of training time and inference speed, which hinders a comprehensive assessment of the model's efficiency.

W2: The authors highlight the consistent depth accuracy of their method as one of the major advantages, providing impressive results in Figure 6. This figure shows their method significantly outperforms all others in outdoor scenes (indicated by the gray background). However, the model fails to achieve state-of-the-art performance on outdoor datasets like KITTI, DIODE Outdoor and ETH3D Outdoor (shown in Table 2 and Table 3). The conflicting results need further explanation from the authors.

W3: In the proposed Campus Depth dataset, the use of CREStereo's predictions as ground truth is questionable. It is unclear whether CREStereo itself suffers from inconsistent accuracy across scenes, raising concerns about the consistency and reliability of its outputs as ground truth.

**Suitability:**

2

---

### Official Review · Reviewer_f8hs · 2024-05-26

**Rating:** 4
**Confidence:** 3

**Summary:**

This paper introduces SM4Depth, a novel algorithm for universal monocular metric depth estimation (MMDE) that addresses the challenges of maintaining consistent accuracy across diverse scenes and reducing reliance on massive training data. The paper introduces an uncut depth dataset, Campus Depth, to evaluate the accuracy consistency across various indoor and outdoor scenes. This dataset allows for the validation of depth accuracy across diverse environments.The proposed method outperforms existing approaches in terms of accuracy and efficiency, demonstrating superior performance with a smaller training dataset.

**Strengths:**

1.	The paper introduces several novel contributions to the field of monocular metric depth estimation (MMDE). The variation-based depth bins proposed in this work offer a fresh perspective on modeling metric scale, addressing the challenges associated with scenes containing large depth gaps.
2.	The proposed SM4Depth algorithm has wide-ranging applications across various multimedia tasks, including video and image editing.
3.	The paper demonstrates a strong motivation to address practical challenges in the MMDE domain, such as inconsistent accuracy across diverse scenes and the reliance on massive training data.

**Limitations:**

1. The paper suggests that by dividing the solution space into sub-spaces and estimating metric bins from various sub-spaces, complexity can be reduced. However, it's unclear how the division of the solution space into sub-spaces is performed and how these sub-spaces are defined.
2. Domain-Aware Bin Estimation  sounds promising in theory, it raises questions about computational efficiency and model scalability.
3. The paper introduces a "divide and conquer" approach to reduce reliance on massive training data by estimating metric bins from various solution sub-spaces. However, the rationale behind this approach is not thoroughly justified.
4. The paper lacks clarity on how incorporating metric bins into each stage of the decoder facilitates better depth range recovery and how it compares to existing refinement decoder structures.
5. It's unclear how representative the selected datasets are of real-world scenarios. For instance, are there specific challenges or biases in the datasets that might affect the performance evaluation?
6. The paper does not discuss potential failure cases or scenarios where the method might underperform.

**Suitability:**

3

---

### Meta-Review · Area_Chair_YFzD · 2024-07-02

**Recommendation:** Accept (Poster)
**Confidence:** 4

**Metareview:**

This paper introduces SM4Depth, a model designed for consistent monocular depth estimation across diverse environments. The model is well motivated and it achieves decent performance, with limited training data. Extensive experimentation demonstrates the effectiveness of the proposed method. Several concerns were raised by the reviewers on lack of consistency in performance across datasets, reliability of the ground truth in the proposed dataset. Some missing analysis and results were also reported. The authors addressed most of these concerns in the rebuttal, however,  reviewers found that some aspects like lack of consistency were not properly addressed. I recommend the authors to add the same in the camera ready version and also include the relevant points from the authors response. Overall, looking at the consensus, I am recommending for acceptance of this paper.